

# Magnetopause as conformal mapping

Yasuhito Narita[1], Simon Toepfer[2], and Daniel Schmid[1]

[1]Space Research Institute, Austrian Academy of Sciences, Schmiedlstr. 6, 8042 Graz, Austria
[2]Institut für Theoretische Physik, Technische Universität Braunschweig, Mendelssohnstr. 3, 38106 Braunschweig, Germany

**Correspondence:** Y. Narita (yasuhito.narita@oeaw.ac.at)

**Abstract.** Magnetopause model is presented as a conformal mapping in the complex plane. The model is an analytic continuation of the power-law damped (or asymptotically elongated) parabolic shape constructed by Shue et al. (J. Geophys. Res., 102, 9497, 1997). The analytic expression of the magnetopause using the conformal mapping opens the door to properly map the magnetopause and magnetosheath coordinates from one model to another.

## 1   Introduction

The magnetopause model proposed by Shue et al. (1997) (hereafter the Shue model) is, to the authors' knowledge, one of the most successful structure models in space science. The Shue model can be given in a simple analytic way by combining a parabolic shape with a power law, and has successfully been tested against the magnetopause of the Earth and the other planets such as Mercury.

Here we report our finding that the magnetopause model can be formulated as a conformal mapping in the complex plane. This mapping preserves local angles. Any analytic function satisfies the conformal (angle-preserving) character in the complex plane as far as there is non-zero derivative. Expression of the magnetopause as a conformal map is ideal when dealing with different magnetopause models.

Our study is motivated to fill the gap between the property of the bow shock models and that of the magnetopause models. The bow shock is often modeled as a conic section (either as a parabola or as a hyperbola) and the analytic expression for the conformal map is known. The magnetopause shape (such as in the Shue model) is, on the other hand, not a conic section, and the existence of conformal mapping remained a question for a long time. We tackle the question by incorporating various conformal mappings.

## 2   Construction of conformal map

We start with the magnetopause model in polar coordinates after Shue et al. (1997),

$$R = R_{\mathrm{mp}} \left( \frac{2}{1 + \cos\theta} \right)^{\alpha},  \tag{1}$$

where $R$ is the radial distance to the planet, $\theta$ is the zenith angle (measured from the planet center), and $\alpha$ is the power index to designate the magnetopause shape in the tail region, e.g., a parabolic shape (corresponds to $\alpha = 1$), an elongated shape (given





by $\alpha = 1/2$), or damped, converging shape ($\alpha < 1/2$). $R_{\mathrm{mp}}$ denotes the magnetopause stand-off distance at the subsolar point.

In our work, we choose $\alpha = 0.5$ which is statistically representative.

By introducing the transformation

$$x = \frac{R}{R_{\mathrm{mp}}} \cos\theta \tag{2}$$

$$y = \frac{R}{R_{\mathrm{mp}}} \sin\theta, \tag{3}$$

the magnetopause location is given in the Cartesian form as

$$x^2 = \frac{4}{4-y^2} - y^2. \tag{4}$$

The derivation of Eq. (4) is shown in Appendix. Note that $x$ and $y$ are normalized to the magnetopause stand-off distance $R_{\mathrm{mp}}$ for simplicity. The magnetopause model (Eq. 4) has the following boundary conditions and asymptotic behavior:

1. The stand-off distance is restored at the subsolar point, i.e., $x = 1$ at $y = 0$.

2. The distance to the planet is $y = \pm\sqrt{2}$ at the terminator ($x = 0$)).

3. The distance to the axis is $y = \pm 2$ when $x \to \infty$.

Now we express Eq. (4) in the complex plane using the variable $z = u + \mathrm{i}v$ so that the magnetopause location is given as

$$f(z) = x + \mathrm{i}y, \tag{5}$$

in other words,

$$x = \mathrm{Re}\,(f(z)) \tag{6}$$

$$y = \mathrm{Im}\,(f(z)) \tag{7}$$

in the Cartesian representation. The complex-valued function $f(z)$ is an extension of the magnetopause location. The magnetopause is restored when choosing $v = v_{\mathrm{mp}} = 1$ (or $v_{\mathrm{mp}}$ to be evaluated as $\sqrt{R_{\mathrm{mp}}}$ when not normalized). The task is thus to find the suitable function $f(z)$.

To our task, we first transform the $y$ coordinates onto the imaginary axis as $\mathrm{i}y$ (where $\mathrm{i}$ is the imaginary unit) so that the

division term in Eq. (4) is formulated from $4/(4-y^2)$ into $4/(4+(\mathrm{i}y)^2)$. Now we perform the analytic continuation of the right hand side of Eq. (4) and replace $\mathrm{i}y$ by $z$.

After some exercises in calculus, we find out that the combination of four sequential conformal mappings is a reasonable analytic continuation of the magnetopause model: ($t_1$) square transformation, ($t_2$) Joukowsky transformation (with shift), ($t_3$) square root transformation, and ($t_4$) scaling and shifting (for the matching with the boundaries). Each transformation is

discussed below.





**(1) Square transformation**

In the first conformal mapping, the square transformation is used with a unit coefficient and no shift. The transformation is expressed as

$$t_1: \quad z \to z^2. \tag{8}$$

The transformation yields the parabolic coordinates as

$$\mathrm{Re}(z^2) = u^2 - v^2 \tag{9}$$
$$\mathrm{Im}(z^2) = 2uv, \tag{10}$$

which can be arranged into a parabolic equation when eliminating $u$ as

$$\mathrm{Re}(z^2) = \frac{\left(\mathrm{Im}(z^2)\right)^2}{4v^2} - v^2. \tag{11}$$

In fact, the parabolic model of magnetopause is introduced by Kobel and Flückiger (1994), which is equivalent to the following transformation

$$t_{\mathrm{K}}: \quad z \to -\frac{1}{2}z^2 + \frac{1}{2}. \tag{12}$$

Here $v = v_{\mathrm{mp}}$ corresponds to the magnetopause location. Figure 1 top left panel displays the mapping of $u = \mathrm{const}$ lines (in gray) $v = \mathrm{const}$ lines (in black) for the transformation $t_1$. The "nose" of magnetopause is located on the negative x side.

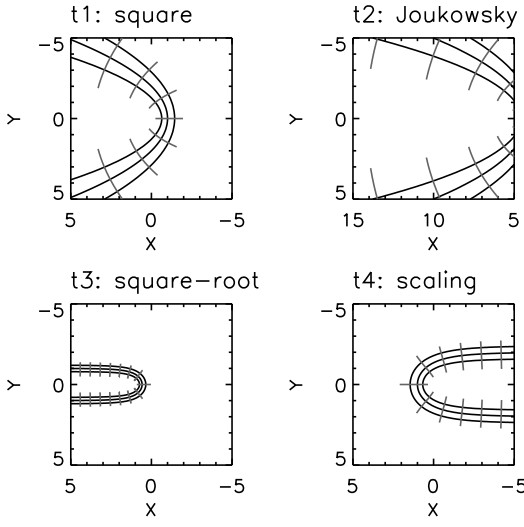

**Figure 1.** Constant $u$ lines (in gray) and constant $v$ lines (in black) for the conformal mappings $t_1$, $t_2$, $t_3$, and $t_4$.





**(2) Shifted Joukowsky transformation**

In the second conformal mapping, the parabolic shape of the mapped curves are stretched using the poles at $z = \pm i2$. The transformation is a variant of Joukowsky transformation, which deforms circles into ellipses (Joukowsky, 1910). We perform the Joukowsky transformation by retaining the pole terms $z^2 + 4$ as

$$t_2: \quad z^2 \to (z^2 + 4) + \frac{4}{z^2 + 4}. \tag{13}$$

Figure 1 top right panel displays the mapping for the transformation $t_2$. The overall structure of $v = \text{const}$ lines still retains the parabolic shape, but the focal point shifts to a larger value of $x$ and the distance from the $x$ axis (the $y = 0$ line) is larger.

**(3) Shifted square root transformation**

In the third conformal mapping, the Joukoswky-transformed function $t_3$ is compared to the magnetopause model (Eq. 4). The comparison yields a subtraction by $4$ and a square-root operation as

$$t_3 \quad : \quad (z^2 + 4) + \frac{4}{z^2 + 4}$$
$$\to \left[ (z^2 + 4) + \frac{4}{z^2 + 4} - 4 \right]^{1/2}. \tag{14}$$

Again, the poles are retained in this transformation. The mapped function has a shape of magnetopause, but the focal point is located in the far tail region and the distance to the magnetopause is smaller than the stand-off distance. Figure 1 bottom left panel displays the mapping for the transformation $t_3$. The tail shape is elongated by this transformation The focal point is 80 moved close to the origin.

**(4) Scaling and shifting**

In the final conformal mapping, the mapping is scaled by a factor $a$ and also shifted by an offset of $f_0$. The transformation reads

$$t_4 \quad : \quad \left[ z^2 + \frac{4}{z^2 + 4} \right]^{1/2}$$
$$\to a \left[ z^2 + \frac{4}{z^2 + 4} \right]^{1/2} + f_0 \tag{15}$$

where the scale factor $a = -2$ is determined by the asymptotic behavior in the tail (distance of $2R_{\text{mp}}$ to the axis) and the shift $f_0 = 1 + 2/\sqrt{3}$ is determined by the stand-off distance at the subsolar point. Combining the four transformations, the scalable magnetopause shape is expressed as a conformal mapping with

$$f(z) \quad = \quad -2 \left[ \frac{4}{z^2 + 4} + z^2 \right]^{1/2} + \left( 1 + \frac{2}{\sqrt{3}} \right). \tag{16}$$

Figure 1 bottom right panel displays the mapping for the transformation $t_4$. The magnetopause nose is flipped to the positive $x$ side. and is scaled to match the magnetopause asymptotic behavior in the tail, and the lines are shifted by $f_0$ along the $x$ axis to meet the stand-off distance.





**Magnetopause location**

The magnetopause location is restored when choosing $v = 1$ in $z = u + iv$. It is also worth noting that the function obtained by

the transformation $t_3$ for $v = 1$ can analytically be evaluated as

$$\frac{4}{z^2+4} + z^2 = \left[ \frac{4(u^2+3)}{(u^2+3)^2+(2u)^2} + (u^2-1) \right] +$$
$$i2u \left[ 1 - \frac{4}{(u^2+3)^2+(2u)^2} \right], \tag{17}$$

which is used to determine the scale factor $a$ and the shift $f_0$ in the transformation $t_4$ by comparing with the square of $f(z)$ as

$$(f - f_0)^2 = a^2 \left( \frac{4}{z^2+4} + z^2 \right). \tag{18}$$

## 3 Applications and limits

### 3.1 Accuracy check

The function $f(z) = x + iy$ using Eq. (16) at $v = 1$ overall reproduces the shape of the Shue model. Figure 2 shows the comparison between the magnetopause model using Eq. (16) and the Shue model. The subsolar point ($x = 1$ at $y = 0$) and the asymptotic behavior ($y \rightarrow \pm 2$ at $x \rightarrow -\infty$) are reproduced, as well. However, it should be noted that the difference occurs from

the Shue model at the terminator ($x = 0$). Our function shows the magnetopause distance at the terminator at $y = \pm 1.3504$, which is slightly underestimating that of the Shue model, $y = \pm\sqrt{2} = \pm 1.4142$. The difference between the two models is about 4.7 %. This mismatch indicates that the analytic continuation is not exact but is of approximate nature. Thus, care should be exercised when working on the magnetopause around the terminator with our conformal mapping.

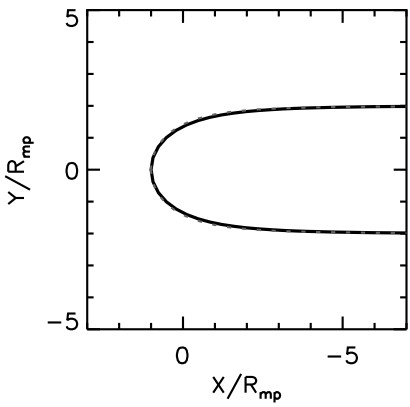

**Figure 2.** Magnetopause location generated by Equation (16) with $v = 1$ in $z = u + iv$ (in black) and the magnetopause model by Shue et al. (1997) (dotted gray). $X$ and $Y$ coordinates are normalized to the magnetopause stand-off distance $R_{\mathrm{mp}}$.



## 3.2 Curvilinear grid generation

The analytic nature of our function (Eq. 16) can be used for the curvilinear grid generation around the magnetopause for various numerical studies. Figure 3 displays the curvilinear grid generated by Eq. 16) for values of $u = \{0.5, 0.7, \cdots, 1.4\}$ (the C-shaped curves) and $v = \{0.0, \pm 0.2, \cdots, \pm 1.4\}$ (radial to the planet or perpendicular to the $X$ axis). The curves of constant $u$ values are orthogonal to that of constant $v$ values. This property comes from the fact that Eq. (16) is an analytic function which is one of the solutions of the Laplace equation. In other words, Eq. (16) solves the Laplace equation for the given magnetopause position (imposed by $v = 1$).

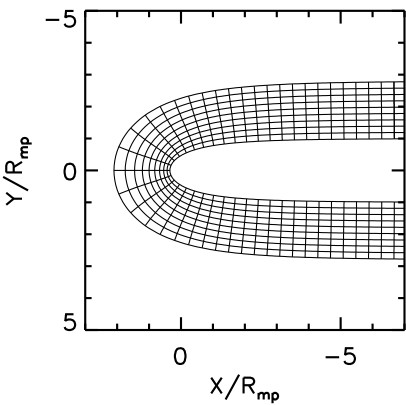

**Figure 3.** Curvilinear grids generated by the conformal map (Eq. 16) around the magnetopause ($v = 1$). The C-shaped curves represent lines of constant $v$ values. The innermost curve corresponds to a line of $v = 0.5$. The $v$ value for the curves are shifted as 0.5, 0.7, $\cdots$, 1.4 (10 curves are shown). The radial curves represent constant $u$ values and the curves are orthogonal to the curves of $v$ values. The subsolar direction $Y = 0$ is given by $u = 0$. The curves are plotted for $u$ values of 0, 0.2, 0.4, $\cdots$, 4.4 (45 curves are shown).


## 3.3 Variation of tail shape

Qualitatively speaking, different tail shapes can also be obtained by generalizing the square-root operation in $t_3$ into a power with the index $\alpha$ as

$$t_3' \quad : \quad (z^2 + 4) + \frac{4}{z^2 + 4}$$

$$\rightarrow \left[ (z^2 + 4) + \frac{4}{z^2 + 4} - 4 \right]^{\alpha}. \tag{19}$$

The magnetopause coordinates are plotted as grids for $\alpha$ of 0.2, 0.4, 0.6, and 0.8 in Fig. 4 by using the scale factor and the shift in $t_4$. A converged tail shape is obtained for $\alpha < 0.5$ and a divergent tail shape for $\alpha > 0.5$, which is in agreement with the Shue model (Eq. 1).



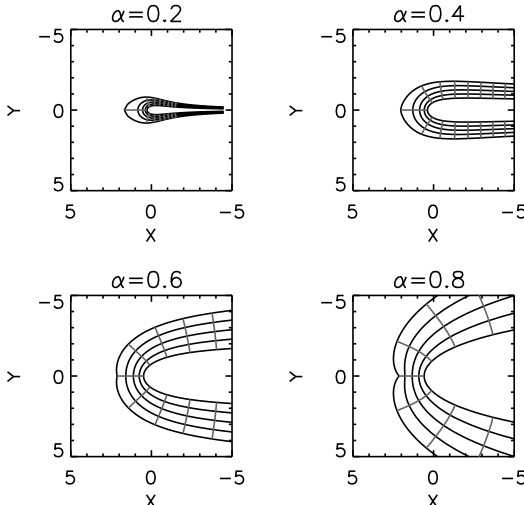

**Figure 4.** Magnetopause grids generated for different values of the power index $\alpha$ in $t_3'$ transformation. Values of $u$ are $0.6$ (innermost C-shaped curve), $0.8, \cdots, 1.4$ (outermost curve).

## 4    Summary and outlook

Conformal mapping is a useful method in the model construction when the axi-symmetry holds and the boundary is modeled in the two-dimensional spatial domain. Our magnetopause model completes the scenario that both dayside boundaries (bow shock and magnetopause) can be modeled by conformal mapping, which opens the door to analytically or semi-analytically map the magnetosheath scalar potential by Kobel and Flückiger (1994) and the set of velocity potential and stream function by Guicking et al. (2012) onto a more realistic magnetosheath domain (cf. Soucek and Escoubet, 2012).

130        The easiest approach of magnetosheath coordinate mapping would be to introduce the transfinite interpolation in the complex plane. Or one could numerically solve the Laplace equation for the given boundaries in order to generate strictly orthogonal curvilinear coordinates.

**Appendix: Magnetopause location in Cartesian**

In the case of $\alpha = 0.5$ the magnetopause position in the Shue model is given by

$$R = R_{\mathrm{mp}}\sqrt{\frac{\ell}{1 + \cos\theta}}, \tag{20}$$

where $\ell = 2$. Equation (20) is transformed sing the conversion rule in Eqs. (2) and (3) into the following normalized form:

$$r = \sqrt{\frac{\ell}{1 + \frac{x}{r}}}, \tag{21}$$





where $r = R/R_{\mathrm{mp}}$. After squaring and exchanging $r$ with $1 + x/r$, Eq. (21) is expressed as

$$1 + \frac{x}{r} = \frac{\ell}{x^2 + y^2}. \tag{22}$$

We compute square of $x/r$ in Eq. (22) and obtain

$$\frac{x^2}{x^2 + y^2} = \left( \frac{\ell}{x^2 + y^2} - 1 \right)^2, \tag{23}$$

which can be arranged into a fourth-order algebraic equation with respect to $y$ as

$$y^4 + x^2 y^2 - 2\ell y^2 - 2\ell x^2 + \ell^2 = 0. \tag{24}$$

The factorized form of Eq. (24) reads

$$(x^2 + y^2)(y^2 - 2\ell) + \ell^2 = 0 \tag{25}$$

Equation (25) delivers the Cartesian representation of the Shue model in a convenient form (Eq. 4).

*Code and data availability.*   No codes or data are used in this work.

*Author contributions.*   All authors listed have made a substantial, direct, and intellectual contribution to the work and approved it for publication.

*Competing interests.*   Conflict of Interest: The authors declare that the research was conducted in the absence of any commercial or financial relationships that could be construed as a potential conflict of interest.





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
