# Peer review of "Magnetopause as conformal mapping"

_Annales Geophysicae, 2022_

## Referee Comment (RC1)

The authors present a technique using conformal mapping in the complex plane to describe various magnetopause models, in a similar way to existing descriptions of bow shock models. This paper has potential for being a useful methodology paper, and I recommend that it be published after a minor revision. My issues (detailed below) mainly concern a somewhat lacking reference to earlier work.

Detailed comments:
* * *
*Abstract: 'Magnetopause model is presented'*

Fix grammar

Also, references in the abstract is usually not recommended.

*line 9: 'The Shue model can be given in a simple analytic way by combining a parabolic shape with a power law, and has successfully been tested against the magnetopause of the Earth and the other planets such as Mercury.'*

Here some relevant references should be given.

*l 12: 'as far as'*

'as long as'

*l 15: 'The bow shock is often modeled as a conic section (either as a parabola or as a hyperbola) and the analytic expression for the conformal map is known.'*

Also here some relevant references would be suitable.

*l 25: 'In our work, we choose alfa = 0.5 which is statistically representative.'*

Again, provide references to substantiate this claim.

*l 35: 'The distance to the axis'*

What axis?

*l 45: 'the division term'*

'the denominator'

*l 47: 'After some exercises in calculus'*

I find this type of 'funny' formulations to just be a distraction. I recommend reformulating to a more neutral formulation,

---

## Author Comment (AC1)

Reply to the referee comments and changes in the revision

> The authors present a technique using conformal mapping in the complex plane
> to describe various magnetopause models, in a similar way to existing descriptions
> of bow shock models. This paper has potential for being a useful methodology paper,
> and I recommend that it be published after a minor revision. My issues (detailed below)
> mainly concern a somewhat lacking reference to earlier work.

Reply

Thank you for the positive evaluation. Suggestions are incorporated in the revision.

> Detailed comments:
> ------------------------
> Abstract: 'Magnetopause model is presented'
> Fix grammar
> Also, references in the abstract is usually not recommended.

Reply

Abstract text was revised by improving grammatics wording, and deleting the reference (page 1, abstract field).

  "An axi-symmetric two-dimensional magnetopause model is constructed by making use of
  the conformal mapping in the complex plane. The model is an analytic continuation of
  the power-law-damped (or asymptotically elongated) parabolic shape. The complex-plane
  expression of the magnetopause opens the door to properly map the magnetopause and
  magnetosheath coordinates from one model to another."

> line 9: 'The Shue model can be given in a simple analytic way by combining
> a parabolic shape with a power law, and has successfully been tested against
> the magnetopause of the Earth and the other planets such as Mercury.'
> Here some relevant references should be given.

Reply

Sure. Agreed. It is (page 1, line 9)

  Winslow, R. M., Anderson, B. J., Johnson, C. L., Slavin, J. A., Korth, H., Purucker,
    M., Baker,   D. N., and Solomon, S. C.: Mercury's magnetopause and bow shock
    from MESSENGER Magnetometer observations, J. Geophys. Res. Space Physics,
    118, 2213–2227, 2013. https://doi.org/10.1002/jgra.50237

> l 12: 'as far as'
> 'as long as'

Reply

Done (page 1, line 12).

> l 15: 'The bow shock is often modeled as a conic section (either as a parabola or
> as a hyperbola) and the analytic expression for the conformal map is known.'
> Also here some relevant references would be suitable.

Reply

Conic section modeling for the bow shock is elaborated by (page 1, line 15):

Cairns, I. H., Fairfield, D. H., Anderson, R. R., Carlton, V. E. H., Paularenas, K. I.,
   and Lazarus, A.: Unusual locations of Earth's bow shock on September 24-25, 1987:
   Mach number effects, J. Geophys. Res., 100, 47–62, 1995.
   https://doi.org/10.1029/94JA01978

Mathematical procedure of the conformal mapping of a conic section is documented in (page 1,
line 16):

Darboux, G.: Leçons sur la théorie générale des surfaces et ses applications géométriques
   du calcul infinitésimal, 1 , Gauthier-Villars, 1887.
   https://gallica.bnf.fr/ark:/12148/bpt6k77831k.image
Encyclopedia of Mathematics, European Mathematical Society, EMS Press, 2020.
    https://encyclopediaofmath.org
Sauer, R., and Szabó, I.: Mathematische Hilfsmittel des Ingenieurs , 1 , Springer, 1967.
   https://link.springer.com/book/9783642949913

> l 25: 'In our work, we choose alfa = 0.5 which is statistically representative.'
> Again, provide references to substantiate this claim.

Reply

Shue et al., 1997 (page 2, line 25).

> l 35: 'The distance to the axis'
> What axis?

Reply

It is "the Sun-Earth axis (or the x-axis in GSE coordinates)" (page 2, line 35).

> l 45: 'the division term'
> 'the denominator'

Reply

Done (page 2, line 45).

> l 47: 'After some exercises in calculus'
> I find this type of 'funny' formulations to just be a distraction. I recommend
> reformulating to a more neutral formulation,

Reply

Agreed. Criticism well taken. I start the sentence simply with "We find out..." (page 2, line 47).

[revised manuscript text omitted]